# Approximating Hierarchical MV-sets for Hierarchical Clustering

**Assaf Glazer**     **Omer Weissbrod**     **Michael Lindenbaum**     **Shaul Markovitch**
Department of Computer Science, Technion - Israel Institute of Technology
{assafgr,omerw,mic,shaulm}@cs.technion.ac.il

## Abstract

The goal of hierarchical clustering is to construct a cluster tree, which can be viewed as the modal structure of a density. For this purpose, we use a convex optimization program that can efficiently estimate a family of hierarchical dense sets in high-dimensional distributions. We further extend existing graph-based methods to approximate the cluster tree of a distribution. By avoiding direct density estimation, our method is able to handle high-dimensional data more efficiently than existing density-based approaches. We present empirical results that demonstrate the superiority of our method over existing ones.

## 1 Introduction

Data clustering is a classic unsupervised learning technique, whose goal is dividing input data into disjoint sets. Standard clustering methods attempt to divide input data into discrete partitions. In Hierarchical clustering, the goal is to find nested partitions of the data. The nested partitions reveal the modal structure of the data density, where clusters are associated with dense regions, separated by relatively sparse ones [27, 13].

Under the nonparametric assumption that the data is sampled i.i.d. from a continuous distribution $F$ with Lebesgue density $f$ in $\mathbb{R}^d$, Hartigan observed that $f$ has a hierarchical structure, called its *cluster tree*. Denote $L_f(c) = \{x \ : \ f(x) \geq c\}$ as the *level set* of $f$ at level $c$. Then, the connected components in $L_f(c)$ are the *high-density clusters* at level $c$, and the collection of all high-density clusters for $c \geq 0$ has a hierarchical structure, where for any two clusters $A$ and $B$, either $A \subseteq B$, $B \subseteq A$, or $A \bigcap B = \emptyset$.

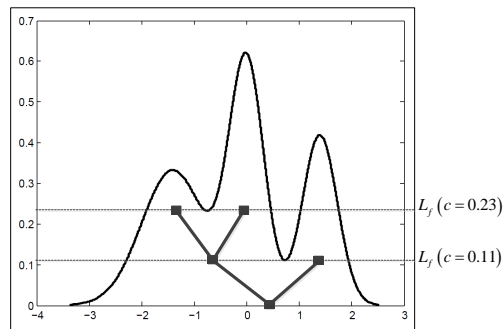

Figure 1: A univariate, tri-modal density function and its corresponding cluster tree are illustrated.

Figure 1 shows a plot of a univariate, tri-modal density function. The cluster tree of the density function is shown on top of the density function. The high-density clusters are nodes in the cluster tree. Leaves are associated with modes in the density function.

Given the density $f$, the cluster tree can be constructed in a straightforward manner via a recursive algorithm [23]. We start by setting the root node with a single cluster containing the entire space, corresponding to $c = 0$. We then recursively increase $c$ until the number of connected components increases, at which point we define a new level of the tree. The process is repeated as long as the number of connected components increases. In Figure 1, for example, the root node has two daughter nodes, which were found at level $c = 0.11$. The next two descendants of the left node were found at level $c = 0.23$.

A common approach for hierarchical clustering is to first use a density estimation method to obtain $f$ [18, 5, 23], and then estimate the cluster tree using the recursive method described above. However, one major drawback in this approach is that a reliable density estimation is hard to obtain, especially in high-dimensional data.

An alternative approach is to estimate the level sets directly, without a separate density estimation step. To do so, we define the minimum volume set (MV-set) at level $\alpha$ as the subset of the input space with the smallest volume and probability mass of at least $\alpha$. MV-sets of a distribution, which are also level sets of the density $f$ (under sufficient regularity conditions), are hierarchical by definition. The well-known One-Class SVM (*OCSVM*) [20] can efficiently find the MV-set at a specified level $\alpha$. A naive approach for finding a hierarchy of MV-sets is to train distinct OCSVMs, one for each MV-set, and enforce hierarchy by intersection operations on the output. However, this solution is not well suited for finding a set of hierarchical MV-sets, because the natural hierarchy of MV-sets is not exploited, leading to a suboptimal solution.

In this study we propose a novel method for constructing cluster trees by directly estimating MV-sets, while guaranteeing convergence to a globally optimum solution. Our method utilizes the *q-One-Class SVM (q-OCSVM)* method [11], which can be regarded as a natural extension of the OCSVM, to jointly find the MV-sets at a set of levels $\{\alpha_i\}$. By avoiding direct density estimation, our method is able to handle high-dimensional data more efficiently than existing density-based approaches. By jointly considering the entire spectrum of desired levels, a globally optimum solution can be found.

We combine this approach with a graph-based heuristic, found to be successful in high-dimensional data [2, 23], for finding high density clusters in the approximated MV-sets. Briefly, we construct a fully connected graph whose nodes correspond to feature vectors, and remove edges between nodes connected by low-density regions. The connected components in the resulting graph correspond to high density clusters.

The advantage of our method is demonstrated empirically on synthetic and real data, including a reconstruction of an evolutionary tree of human populations using the high-dimensional 1000 genomes dataset.

## 2   Background

Our novel method for hierarchical clustering belongs to a family of non-parametric clustering methods. Unlike parametric methods, which assume that each group $i$ is associated with a density $f_i$ belonging to some family of parametric densities, non-parametric methods assume that each group is associated with modes of a density $f$ [27]. Non-parametric methods aim to reveal the modal structure of $f$ [13, 28, 14].

Hierarchical clustering methods can be divided into agglomerative (bottom up) and divisive (top down) methods. Agglomerative methods (e.g. single-linkage) start with $n$ singleton clusters, one for each training feature vector, and work by iteratively linking two closest clusters. Divisive methods, on the other hand, start with all feature vectors in a single cluster and recursively divide clusters into smaller sub-clusters.

While single-linkage was found, in theory, to have better stability and convergence properties in comparison to average-linkage and complete-linkage [4], it is frequently criticized by practitioners due to the *chaining effect*. Single-linkage ignores the density of feature vectors in clusters, and thus may erroneously connect two modes (clusters) with a few feature vectors connecting them, that is, a 'chain" of feature vectors.

Wishart [27] suggested overcoming this effect by conducting a *one-level analysis* of the data. The idea is to estimate a specific level set of the data density ($L_f(c)$), and to remove noisy features

outside this level that could otherwise lead to the chaining effect. The connected components left in $L_f(c)$ are the clusters; expansions of this idea can be found in [9, 26, 6, 3]. Indeed, this analysis is more resistant to the chaining effect. However, one of its major drawbacks is that no single level set can reveal all the modes of the density. Therefore, various studies have proposed estimating the entire hierarchical structure of the data (the cluster tree) using density estimates [13, 1, 22, 18, 5, 23, 17, 19]. These methods are considered as divisive hierarchical clustering methods, as they start by associating all feature vectors to the root node, which is then recursively divided to sub-clusters by incrementally exploring level sets of denser regions. Our proposed method belongs to this group of divisive methods.

Stuetzle [22] used the nearest neighbor density estimate to construct the cluster tree and pointed out its connection to single-linkage clustering. Kernel density estimates were used in other studies [23, 19]. The *bisecting K-means (BiKMean)* method is another divisive method that was found to work effectively in cluster analysis [16], although it provides no theoretical guarantee for finding the correct cluster tree of the underlying density.

Hierarchical clustering methods can be used as an exploration tool for data understanding [16]. The nonparametric assumption, by which density modes correspond to homogenous feature vectors with respect to their class labels, can be used to infer the hierarchical class structure of the data [15]. An implicit assumption is that the closer two feature vectors are, the less likely they will be to have different class labels. Interestingly, this assumption, which does not necessarily hold for all distributions, is being discussed lately in the context of hierarchical sampling methods for active learning [8, 7, 25], where the correctness of such a hierarchical modeling approach is said to depend on the "Probabilistic Lipschitzness" assumption about the data distribution.

## 3 Approximating MV-sets for Hierarchical Clustering

Our proposed method consists of (a) estimating MV-sets using the *q-OCSVM* method; (b) using a graph-based method for finding a hierarchy of high density regions in the MV-sets, and (c) constructing a cluster tree using these regions. These stages are described in detail below.

### 3.1 Estimating MV-Sets

We begin by briefly describing the One-Class SVM (OCSVM) method. Let $\mathcal{X} = \{x_1, \ldots, x_n\}$ be a set of feature vectors sampled i.i.d. with respect to $F$. The function $f_C$ returned by the *OCSVM* algorithm is specified by the solution of this quadratic program:

$$\min_{w \in \mathcal{F}, \xi \in \mathbb{R}^n, \rho \in \mathbb{R}} \frac{1}{2}||w||^2 - \rho + \frac{1}{\nu n} \sum_i \xi_i,$$
$$s.t. \ \ (w \cdot \Phi(x_i)) \geq \rho - \xi_i, \ \xi_i \geq 0,$$

(1)

where $\xi$ is a vector of the slack variables. Recall that all training examples $x_i$ for which $(w \cdot \Phi(x)) - \rho \leq 0$ are called *support vectors (SVs)*. Outliers are referred to as examples that strictly satisfy $(w \cdot \Phi(x)) - \rho < 0$. By solving the program for $\nu = 1 - \alpha$, we can use the *OCSVM* to approximate the MV-set $C(\alpha)$.

Let $0 < \alpha_1 < \alpha_2, \ldots, < \alpha_q < 1$ be a sequence of $q$ quantiles. The *q-OCSVM* method generalizes the *OCSVM* algorithm for approximating a set of MV-sets $\{C_1, \ldots, C_q\}$ such that a hierarchy constraint $C_i \subseteq C_j$ is satisfied for $i < j$. Given $\mathcal{X}$, the *q-OCSVM* algorithm solves this primal program:

$$\min_{w, \xi_j, \rho_j} \frac{q}{2}||w||^2 - \sum_{j=1}^q \rho_j + \sum_{j=1}^q \frac{1}{\nu_j n} \sum_i \xi_{j,i}$$
$$s.t. \ \ (w \cdot \Phi(x_i)) \geq \rho_j - \xi_{j,i}, \ \xi_{j,i} \geq 0, \ j \in [q], i \in [n],$$

(2)

where $\nu_j = 1 - \alpha_j$. This program generalizes Equation (1) to the case of finding multiple, parallel half-space decision functions by searching for a global minimum over their sum of objective functions: the coupling between $q$ half-spaces is done by summing $q$ *OCSVM* programs, while forcing these programs to share the same $w$. As a result, the $q$ half-spaces in the solution of Equation (2) differ only by their bias terms, and are thus parallel to each other. This program is convex, and thus a global minimum can be found in polynomial time.

Glazer et al. [11] proves that the $q$-OCSVM algorithm can be used to approximate the MV-sets of a distribution.

### 3.1.1 Generalizing $q$-*OCSVM* for Finding an Infinite Number of Approximated MV-sets

The $q$-*OCSVM* finds a finite number of $q$ approximated MV-sets, which capture the overall structure of the cluster tree. However, in order to better resolve differences in density levels between data points, we would like the solution to be extended for defining an infinite number of hierarchical sets.

Our approach for doing so relies on the parallelism property of the approximated MV-sets in the $q$-*OCSVM* solution. An infinite number of approximated MV-sets are associated with separating hyperplanes in $\mathcal{F}$ that are parallel to the $q$ hyperplanes in the $q$-*OCSVM* solution. Note that every projected feature vector $\Phi(x)$ lies on a unique separating hyperplane that is parallel to the $q$ hyperplanes defined by the solution, and the distance $dis(x) = (w \cdot \Phi(x)) - \rho$ is sufficient to determine whether $x$ is located inside each of the approximated MV-sets.

We would like to know the probability mass associated with each of the infinite hyperplanes. For this purpose, we could similarly estimate the expected probability mass of the approximated MV-set defined for any $x \in \mathbb{R}^d$. When $\Phi(x)$ lies strictly on one of the $i \in [q]$ hyperplanes, then $x$ is considered as lying on the boundary of the set approximating $C(\alpha_i)$. When $\Phi(x)$ does not satisfy this condition, we use a linear interpolation to define $\alpha$ for its corresponding approximated MV-set: Let $\rho_i, \rho_{i+1}$ be the bias terms associated with the $i$ and $i + 1$ approximated MV-sets that satisfy $\rho_i > (w \cdot \Phi(x)) > \rho_{i+1}$. Then we linearly interpolate $(w \cdot \Phi(x))$ along the $[\rho_{i+1}, \rho_i]$ interval for an intermediate $\alpha \in (\alpha_i, \alpha_{i+1})$. For the completion of the definition, we set $\rho_0 = \max_{x \in \mathcal{X}} (w \cdot \Phi(x))$ and $\rho_{q+1} = \min_{x \in \mathcal{X}} (w \cdot \Phi(x))$.

## 3.2 Finding a Hierarchy of High-Density Regions

To find a hierarchy of high density regions, we adopt a graph-based approach. We construct a fully-connected graph whose nodes correspond to feature vectors, and remove edges between nodes separated by low-density regions. The connected components in the resulting graph correspond to high density regions. The method proceeds as follows.

Let $\alpha(x)$ be the expected probability mass of the approximated MV-set defined by $x$. Let $\alpha_{i,s}$ be the maximal value of $\alpha(x)$ over the line segment connecting the feature vectors $x_i$ and $x_s$ in $\mathcal{X}$:

$$\alpha_{i,s} = \max_{t \in [0,1]} \alpha(tx_i + (1 - t)x_s). \tag{3}$$

Let $G$ be a complete graph between pairs of feature vectors in $\mathcal{X}$ with edges equal to $\alpha_{i,s}$ [1]. High density clusters at level $\alpha$ are defined as the connected components in the graph $G(\alpha)$ induced by removing edges from $G$ with $\alpha_{i,s} > \alpha$. This method guarantees that two feature vectors in the same cluster of the approximated MV-set at level $\alpha$ would surely lie in the same connected component in $G(\alpha)$. However, the opposite would not necessary hold — when $\alpha_{i,s} > \alpha$ and a curve connecting $x_i$ and $x_s$ exists in the cluster, $x_i$ and $x_s$ might erroneously be found in different connected components. Nevertheless, it was empirically shown that erroneous splits of clusters are rare if the density function is smooth [23].

One way to implement this method for finding high density clusters is to iteratively find connected components in $G(\alpha)$, when at each iteration $\alpha$ is incrementally increased (starting from $\alpha = 0$), until all the clusters are found. However, [23] observed that we can simplify this method by working only on the graph $G$ and its minimal spanning tree $T$. Consequently, we can compute a hierarchy of high-density regions in two steps: First, construct $G$ and its minimal spanning tree $T$. Then, remove edges from $T$ in descending order of their weights such that the connected components left after removing an edge with weight $\alpha$ correspond to a high density cluster at level $\alpha$. Connected components with a single feature vector are treated as outliers and removed.

### 3.3 Constructing a Cluster Tree

The hierarchy resulting from the procedure described above does not form a full partition of the data, as in each edge removal step a fraction of the data is left outside the newly formed high density clusters. To construct a full partition, feature vectors left outside at each step are assigned to their nearest cluster. Additionally, when a cluster is split into sub-clusters, all its assigned feature vectors are assigned to one of the new sub-clusters.

The choice of kernel width has a strong effect on the resulting cluster tree. On the one hand, a large bandwidth may lead to the inner products induced by the kernel function being constant; that is, many examples in the train data are projected to the same point in $\mathcal{F}$. Hence, the approximated MV-sets could eventually be equal, resulting in a cluster tree with a single node. On the other hand, a small bandwidth may lead to the inner products becoming closer to zero; that is, points in $\mathcal{F}$ tend to lie on orthogonal axes, resulting in a cluster tree with many branches and leaves.

We believe that the best approach for choosing the correct bandwidth is based on the number of modes that we expect to find for the density function. By using a grid search over possible $\gamma$ values, we can choose the bandwidth that results in a cluster tree in which the expected number of modes is the same as the number we expect.

## 4 Empirical Analysis

We evaluate our hierarchical clustering method on synthetic and real data. While the quality of an estimated cluster tree for the synthetic data can be evaluated by comparing the resulting tree with the true modal structure of the density, alternative quality measures are required to estimate the efficiency of hierarchical clustering methods on high-dimensional data when the density is unknown. In the following section we introduce our proposed measure.

### 4.1 The Quality Measure

One prominent measure is the $F$-measure, which was extended by [16] to evaluate the quality of estimated cluster trees. Recall that classes refer to the true (unobserved) class assignment of the observed vectors, whereas clusters refer to their tree-assigned partition. For a cluster $j$ and class $i$, define $n_{i,j}$ as the number of feature vectors of class $i$ in cluster $j$, and $n_i, n_j$ as the number of feature vectors associated with class $i$ and with cluster $j$, respectively. The $F$-measure for cluster $j$ and class $i$ is given by $F_{i,j} = \frac{2*Recall_{i,j}*Precision_{i,j}}{Recall_{i,j}+Precision_{i,j}}$, where $Recall_{i,j} = \frac{n_{i,j}}{n_i}$ and $Precision_{i,j} = \frac{n_{i,j}}{n_j}$. The $F$-measure for the cluster tree is

$$F = \sum_i \frac{n_i}{n} \max_j \{F_{i,j}\}. \tag{4}$$

The $F$-measure was found to be a useful tool for the evaluation of hierarchical clustering methods [21], as it quantifies how well we could extract $k$ clusters, one for each class, that are relatively "pure" and large enough with respect to their associated class. However, we found it difficult to use this measure directly in our analysis, because it appears to prefer overfitted trees, with a large number of spurious clusters.

We suggest correcting this bias via cross-validation. We split the data $\mathcal{X}$ into two equal-sized train and test sets, and construct a tree using the train set. Test examples are recursively assigned to clusters in the tree in a top-down manner, and the $F$-measure is calculated according to the resulting tree. When analytical boundaries of clusters in the tree are not available (such as in our method), we recursively assign each test example in a cluster to the sub-cluster containing its nearest neighbor in the train set, using Euclidean distance.

### 4.2 Reference Methods

We compare our method with methods for density estimation, that can also be used to construct a graph $G$. For this purpose, since $f(x)$ is used instead of $\alpha(x)$, we had to adjust the way we construct

$G$ and $T$ [2]. A *kernel density estimator (KDE)* and *nearest neighbor density estimator (NNE)*, similar to the one used by [23], are used as competing methods. In addition, we compare our method with the *bisecting K-means (BiKMean)* method [21] for hierarchical clustering.

## 4.3   Experiments with Synthetic Data

We run our hierarchical clustering method on data sampled from a synthetic, two-dimensional, trimodal distribution. This distribution is defined by a 3-Gaussian mixture distribution. 20 i.i.d. points were sampled for training our $q$-*OCSVM* method, with $\alpha_1 = 0.25, \alpha_2 = 0.5, \alpha_3 = 0.75$ (3-quantiles), and with a bandwidth $\gamma$, which results in a cluster tree with 3 modes. The left side of Figure 2 shows the data sampled, and the 3 approximated hierarchical MV-sets. The resulting 3-modes cluster tree is shown in the right side of Figure 2.

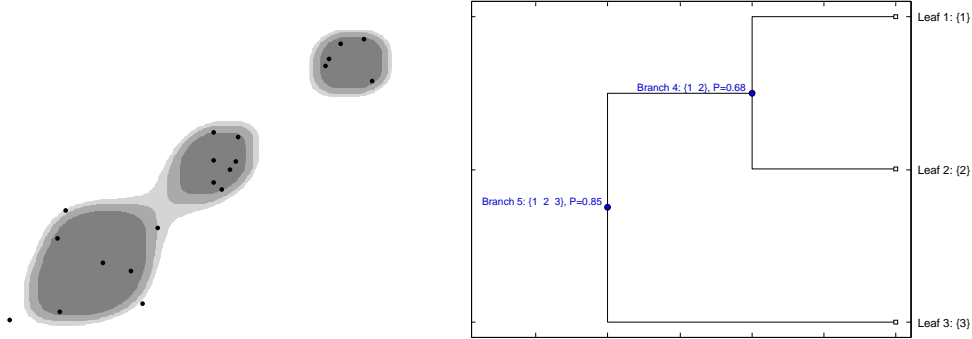

Figure 2: *Left:* Data sampled for training our $q$-*OCSVM* method and the 3 approximated MV-sets; *Right:* The cluster tree estimated from the synthetic data. The most frequent label in each mode, denoted in curly brackets next to each leaf, defines the label of the mode. Branches are labeled with the probability mass associated with their level set.

We used our proposed and reference method on the data to obtain cluster trees with different numbers of modes (leaves). The number of modes can be tweaked by changing the value of $\gamma$ for the $q$-*OCSVM* and *KDE* methods, and by pruning nodes of small size for the *NNE* and *BiKMean* methods. 20 test examples were i.i.d. sampled from the same distribution to estimate the resulting $F$-measures. The left side of Figure 3 shows the $F$-measure for each method in terms of changes in the number of modes in the resulting tree. For all methods, the $F$-measure is bounded by 0.8 as long as the number of modes is greater than 3, correctly suggesting the presence of 3 modes for the data.

## 4.4   The olive oil dataset

The olive oil dataset [10] consists of 572 olive oil examples, with 8 features each, from 3 regions in Italy ($R1, R2, R3$), each one further divided into 3 sub-areas. The right side of Figure 3 shows the $F$-measure for each method in terms of changes in the number of modes in the tree. The $q$-*OCSVM* method dominates the other three methods when the number of modes is higher than 5, with an average $F = 0.62$, while its best competitor (*KDE*) has an average $F = 0.55$.

It can be seen that the variability of the $F$-measure plots is higher for the $q$-*OCSVM* and *KDE* methods than for the *BiKMeans* and *NNE* methods. This is a consequence of the fact that the structure of unpruned nodes remains the same for the *BiKMeans* and *NNE* methods, whereas different $\gamma$ values may lead to different tree structures for the $q$-*OCSVM* and *KDE* methods.

The cluster trees estimated using the $q$-*OCSVM* and *KDE* methods are shown in Figure 4. For each method, we chose to show the cluster tree with the smallest number of modes with leaves corresponding to all 8 labels. The $q$-*OCSVM* method groups leaves associated with the 8 areas into 3 clusters, which perfectly corresponds to the hierarchical structure of the labels. In contrast, modes estimated using the *KDE* method cannot be grouped into 3 homogeneous clusters.

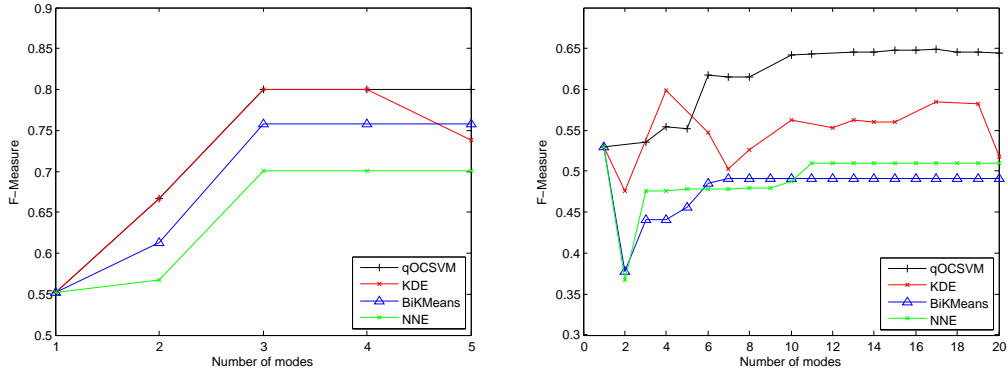

Figure 3: *Left:* The $F$-measures of each method are plotted in terms of the number of modes in the estimated cluster trees. The $F$-measures are calculated using the synthetic test data; *Right:* $F$-measure for the olive oil dataset, calculated using $286$ test examples, is shown in terms of the number of modes in the cluster tree.

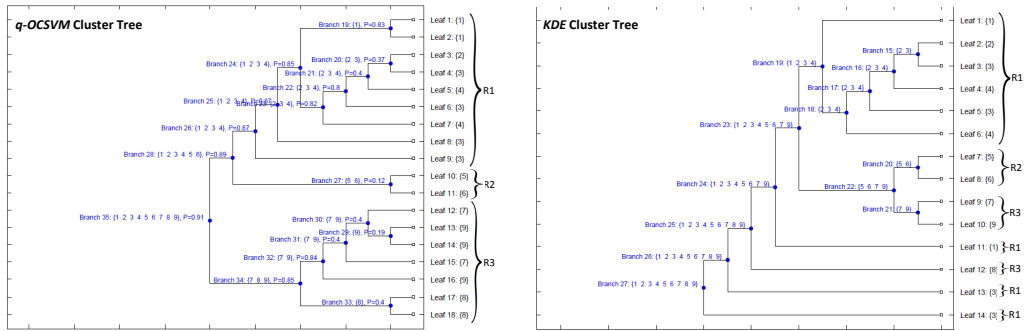

Figure 4: *Left:* Cluster tree for the olive oil data estimated with $q$-$OCSVM$; *Right:* Cluster tree for the olive oil data estimated with *KDE*.

One prominent advantage of our method is that we can use the estimated probability mass of branches in the tree to better understand the modal structure of the data. For instance, we can learn from Figure 4 that the R2 cluster is found in a relatively sparse MV-set at level $0.89$, while its two nodes are found in a much denser MV-set at level $0.12$. Probability masses for high density clusters can also be estimated using the *KDE* method, but unlike our method, theoretical guarantees are not provided.

## 4.5 The $1000$ genomes dataset

We have also evaluated our method on the $1000$ genomes dataset [24]. Hierarchical clustering approaches naturally arise in genetic population studies, as they can reconstruct trees that describe evolutionary history and are often the first step in evolutionary studies [12]. The reconstruction of population structure is also crucial for genetic mapping studies, which search for genetic factors underlying genetic diseases.

In this experiment we evaluated our method's capability to reconstruct the evolutionary history of populations represented in the $1000$ genomes dataset, which consists of whole genome sequences of $1,092$ human individuals from 14 distinct populations. We used a trinary representation wherein each individual is represented as a vector of features corresponding to 0,1 or 2. Every feature represents a known genetic variation (with respect to the standard human reference genome [3]), where the number indicates the number of varied genome copies. We used data processed by the $1000$ Genomes Consortium, which initially contained $2.25$ million variations. To reduce dimensionality, we used the $1,000$ features that had the highest information gain with respect to the populations. We excluded from the analysis highly genetically admixed populations (Colombian, Mexican and Puerto

Rican ancestry), because the evolutionary history of admixed populations cannot be represented by a tree. After exclusion, $911$ individuals remained in the analysis.

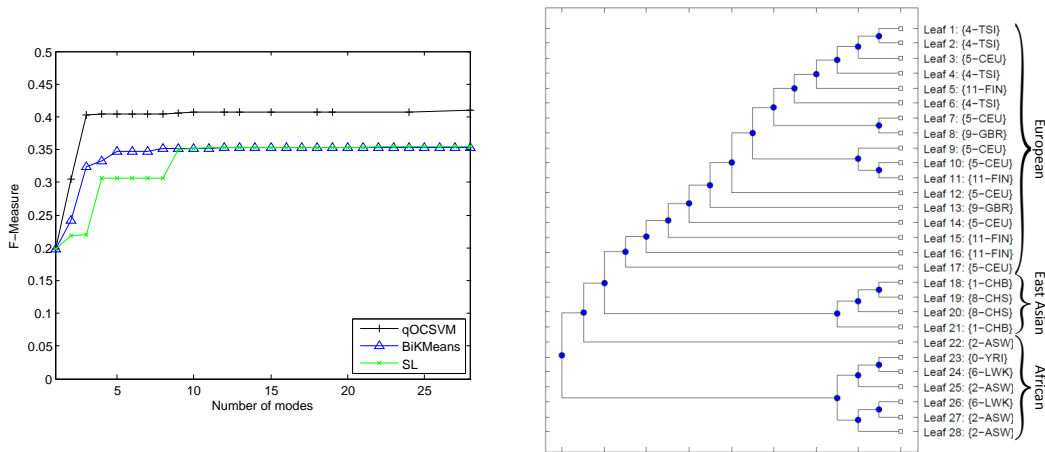

Figure 5: *Left: $F$-measure for the $1000$ genomes dataset, calculated using $455$ test examples; Right:* Cluster tree for the $1000$ genomes data estimated with $q$-*OCSVM*. The labels are GBR (British in England and Scotland), TSI (Toscani in Italia), CEU (Utah Residents with Northern and Western European ancestry), FIN (Finnish in Finland), CHB (Han Chinese in Bejing, China), CHS (Southern Han Chinese), ASW (Americans of African Ancestry in SW USA), YRI (Yoruba in Ibadan, Nigera), and LWK (Luhya in Webuye, Kenya).

The left side of Figure 5 shows that $q$-*OCSVM* dominates the other methods for every number of modes tested, demonstrating its superiority in high dimensional settings. Namely, it achieves an $F$-measure of $0.4$ for $>2$ modes, whereas competing methods obtain an $F$-measure of $0.35$. *KDE* was not evaluated as it is not applicable due to the high data dimensionality.

To obtain a meaningful tree, we increased the number of modes until leaves corresponding to all three major human population groups (African, East Asian and European) represented in the dataset appeared. The tree obtained by using $28$ modes is shown in the right side of Figure 5, indicating that $q$-*OCSVM* clustering successfully distinguishes between these three population groups. Additionally, it corresponds with the well-established theory that a divergence of a single ancestral population into African and Eurasian populations took place in the distant past, and that Eurasians diverged into East Asian and European populations at a later time [12]. The larger number of leaves representing European populations may result from the larger number of European individuals and populations in the $1000$ genomes dataset.

## 5 Discussion

In this research we use the $q$-*OCSVM* method as a plug-in method for hierarchical clustering in high-dimensional distributions. The $q$-*OCSVM* method estimates the level sets (MV-sets) directly without a density estimation step. Therefore, we expect to achieve more accurate results than approaches based on density estimation. Furthermore, since we know $\alpha$ for each approximated MV-set, we believe our solution would be more interpretable and informative than a solution provided by a density estimation-based method.

## Footnotes

[1] We calculated $\alpha_{i,s}$ in $G$ by checking the $\alpha(x)$ values for 20 points sampled from the line segment between $x_i$ and $x_s$. The same approach was also used by [2] and [23].

[2]When a density estimator $f$ is used, $p_{i,s} = \min_{t \in [0,1]} p(tf(x_i) + (1-t)f(x_s))$ are set to be the edge weights, $G(c)$ is induced by removing edges from $G$ with $p_{i,s} < c$, and $T$ is defined as the maximal spanning tree of $G$ (instead of the minimal).

[3]http://genomereference.org

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
