[Reviews · NeurIPS 2014]

Submitted by Assigned_Reviewer_9

+++update+++
The authors addressed my main concerns in their reply; that is why I have raised my score.

The authors propose a novel approach for hierarchical clustering of multivariate data. They construct cluster trees by estimating minimum volume sets using the q-One-Class SVM, and evaluate their method on a synthetic data set and two real word applications.

While their new method seems to perform better than other approaches based on density estimation, I am not convinced by the benefits in practical applicability as the authors did not compare their method to the most commonly used hierarchical clustering techniques (agglomerative clustering with average linkage/ward).

Minor comment: Rather than splitting their data once in a training and test set, the authors should perform 10-fold/5-fold cross-validation for a more reliable estimation of the generalizability of their method.
Summary: The authors present a good paper which unfortunately lacks comparison with the most commonly used alternative method.

Submitted by Assigned_Reviewer_26

The paper introduces a new method for estimating cluster trees by direct estimation of the MV-sets rather than separate density estimation and recursive estimation of the tree using level sets. This avoids the issues inherent in producing stable high-dimensional density estimates. The method uses a graph cutting technique (with the full dataset being the graph nodes) to produce a set of high-density clusters.

Essentially the q-OCSVM method is used to create a different method for specifying the graph edge weights between observations. After this point the algorithm largely follows the cluster tree estimation method set out by Stuetzle and Nugent. The issue of bandwidth selection is an important one and the authors mention it only briefly. Their suggestion of specifying the bandwidth which gives the number of modes that the user would expect does allow for expert knowledge to be incorporated into the procedure which is an advantage (and could be emphasised more in the paper) but only when such knowledge is available. They suggest no alternative for cases where a more objective solution might be required.

The simulations are acceptable and the methods chosen for comparison are reasonable.

Quality: The motivation for the new method is clear and the analysis/results sections are reasonable. It is an interesting approach that could be of use.

Clarity: The paper is very well written, with a good summary of the background of all elements of the approach.

Originality: This method is the application of an already developed method for MV estimation to an existing graph-based approach to building cluster trees. The combination is novel but the different elements are not.

Significance: The results seem to suggest that the new graph edge weight metric has merit but there are outstanding issues with the choice of bandwidth that I suspect will limit the impact of this work on practitioners.

Small details:

If you have space I would suggest emphasising the advantage of this method over existing density-estimation based methods in high-dimensional datasets in the abstract.

The authors give a very nice summary of the concept of a (non-parametric) cluster tree and level sets. Figure 1 level set lines should really have (light) vertical lines dropping to x-axis at points of intersection with the density to illustrate the level sets properly.

Line 082: The phrase "feature vectors" (and "features") seems a little confusing here and elsewhere. Perhaps "observation vectors" would be better. Since "features" often refers to variables.

Line 232, expected is used too many times. Perhaps change the second to the "number specified/recommended by the user".

The description of the quality measure in section 4.1 needs more detail. Better clarifying what it meant by cluster and class here would improve the readability of this part. Are clusters defined just for a single level or over all levels? Similarly are classes defined over a tree or a single level? Presumably clusters refers to the partitions resulting from the tree estimated by the method in the paper?

There is an issue in formatting with the caption of Figure 3 overlapping the line on the bottom of page 6.

There needs to be a line to space things out between the bottom of the captions for Figures 4 and 5 and the main text.

Line 370 "1, 092" should be "1,092"
Summary: The main contribution of this paper seems to be applying q-OCSVM to create graph weights to be used in the cluster tree method from Stuetzle and Nugent. The method performs well in simulations in comparison to other similar methods. The choice of bandwidth and related selection of number of nodes is not well dealt with.

Submitted by Assigned_Reviewer_33

This paper applies q-Quantile Estimators for High-Dimensional Distributions to the task of hierarchical clustering. q-Quantile Estimators are an extension of one-class SVMs to jointly find minimum volume sets at various levels. As these q-quantiles are nested, it is a natural step to use them for hierarchical clustering.

Quality: This paper is technically sound and complete. I like the cross-validation methodology. I don't like the grid search method for finding gamma. It is similar to cutting a linkage based hierarchical clustering to give the "right" number of clusters. Could the authors draw on the literature for deciding the cut level rather than where "the expected number of nodes is the same as the number we expect"?
(The authors have responded to this point noting that it is a "fundamental problem". They choose to restrict their method to those for which the user can supply the right number of modes; that's ok but I urge them to be very explicit and clear about this in the revised version if accepted.)

Also, for the 1000 genomes data, is Leaf 22 is the wrong / it's own cluster? (The authors did not address this point during in their rebuttal.)

Clarity: The paper is clear for the most part. Familiarity with SVM terminology is assumed in places. e.g. in 3.1 notation and variables are introduced without explanation.

Originality: The paper is original as it applies an extension to a new technique.

Significance: This is perhaps small. The q-quantile estimator looks a natural method for hierarchical clustering so perhaps it doesn't take a lot of imagination to use it for this. On the other hand, it appears to be a powerful and new method.
Summary: This is a well written paper that proposes a small but useful extension / application to a NIPS paper from last year.
Author Feedback
Author rebuttal: We thank the reviewers for their helpful comments, and address their comments below.

Bandwidth estimation: Parameter tuning for kernel methods, in particular for OCSVMs, is an ongoing research area. Unlike binary classification tasks, negative examples are not available to estimate the optimality of the solution. A common practice for setting the kernel width is using a fixed width, divided by the number of features (Chih-Chung Chang and Chih-Jen Lin. LIBSVM: a library for support vector machines, 2011; Azmandian et-al, Local Kernel Density Ratio-Based Feature Selection for Outlier Detection, JMLR, 2012). Another heuristic is to set \sigma to the median distance between points in the training set (Gretton et al., A Kernel Two-Sample Test, JMLR, 2012).

In this paper we do not provide a solution for this fundamental problem. Instead, we take a different approach, which is frequently used when clustering problems are considered (Rui Xu et al., Survey of Clustering Algorithms, 2005), of asking the number of modes to be provided by the user. As described in our paper, there is a strong connection between the two problems of finding the correct bandwidth and finding the right number of modes in densities; note that the later problem is also considered as a fundamental issue (Davé et al., Robust clustering methods: A unified view, IEEE Trans. Fuzzy Syst., 1997). A discussion about this topic will be added to the paper.

2. Lack of comparison with agglomerative clustering: In this paper we took the same approach as in Stuetzle (2010) and focused on a comparison of divisive hierarchical clustering methods, which were proven as superior to agglomerative methods in high-dimensional settings (Bjornar et al., Fast and effective text mining using linear-time document clustering, SIGKDD, 1999). However, it is interesting to observe that the divisive NNE method we evaluated is equivalent to a single-linkage clustering method (Stutzle, 2003), so in fact we compared our method also to an agglomerative method. (Note that a comment about this similarity can be found in line 117.) We believe a comparison to average-linkage method is not essential due to the superior stability and convergence properties of the single-linkage method over average/full-linkage methods (Carlsson et al., Characterization, stability and convergence of hierar- chical clustering methods, JMLR, 2010.); A comment on this issue can be found in line 102.

3. Definition of clusters and classes: In Section 4.1, classes refer to the true (unobserved) class assignment of the observed vectors, whereas clusters refer to their tree-assigned partition. For the purposes of the F-measure computation, we refer to each leaf as a cluster. When recursively assigning test samples to the learned tree, we treat each internal node as a cluster. We will clarify these concepts in a camera-ready version.

4. Typos, phrasing suggestions and visual inconsistencies: We thank the reviewers for noticing these, and will correct these in a camera-ready version.